# Pathogenicity of Asymptomatically Residing *Fusarium* Species in Non-Gramineous Plants and Weeds to Spring Wheat under Greenhouse Conditions

**DOI:** 10.3390/pathogens11121467

**Published:** 2022-12-04

**Authors:** Evelina Zavtrikovienė, Andrii Gorash, Gražina Kadžienė, Neringa Matelionienė, Skaidrė Supronienė

**Affiliations:** 1Microbiology Laboratory, Institute of Agriculture, Lithuanian Research Centre for Agriculture and Forestry, Instituto al. 1, Akademija, LT-58344 Kėdainiai, Lithuania; 2Department of Cereal Breeding, Institute of Agriculture, Lithuanian Research Centre for Agriculture and Forestry, Instituto al. 1, Akademija, LT-58344 Kėdainiai, Lithuania; 3Department of Soil and Crop Management, Institute of Agriculture, Lithuanian Research Centre for Agriculture and Forestry, Instituto al. 1, Akademija, LT-58344 Kėdainiai, Lithuania

**Keywords:** asymptomatic alternative hosts, Fusarium head blight, grain yield losses

## Abstract

Despite significant efforts in recent decades to combat Fusarium head blight (FHB), this disease remains one of the most important and widely studied diseases of wheat and other cereal plants. To date, studies have focused on small grain cereals as hostplants for these pathogens, but it was recently discovered that asymptomatic non-gramineous plants and weeds can serve as alternative sources of fungi associated with FHB. The aim of this study was to evaluate the pathogenicity of *Fusarium avenaceum*, *F. culmorum*, *F. graminearum* and *F. sporotrichioides* isolated from non-gramineous plants and weed species to spring wheat under greenhouse conditions. A total of 91 *Fusarium* isolates, including 45 from weeds and 46 from non-gramineous plants were floret inoculated at mid anthesis. The FHB incidence and severity (%) of inoculated heads and the area under the disease progress curve (AUDPC) were calculated. To determine yield losses, the weight of 1000 grains (TGW) was evaluated. Results of the research showed that FHB severity (%) values in *Fusarium* spp.-inoculated heads from non-gramineous plants varied from 9.3% to 69.6% and AUDPC values ranged from 161.5% to 1044.6%. TGW was most significantly reduced by the *F. culmorum* isolates BN26r and BN39fl from *Brassica napus* and isolates BV15.1l and BV142.1pe from *Beta vulgaris* (37%, 30%, 28.8% and 31.8% respectively, compared to the water control). In *Fusarium*-inoculated heads from weeds, FHB severity values ranged from 6.2% to 81.0% and AUDPC values varied from 134.2% to 1206.6%. TGW was most significantly decreased by CBP1401r isolate from *Capsella bursa-pastoris* (52%). The study results suggest that the pathogenicity of *Fusarium* species isolated from different hosts to wheat more strongly depends on the *Fusarium* species and strain than the hostplant. Under greenhouse conditions, *F. culmorum* strain groups obtained from weeds, non-gramineous plants and *Triticum* were more pathogenic to wheat than the water control and other *Fusarium* species.

## 1. Introduction

Fusarium head blight (FHB), which is primarily caused by several *Fusarium* species and their complexes, is a disease that affects wheat and other small-grain cereals [1,2,3,4,5]. Although FHB has been heavily combated in recent years, it continues to be one of the most globally significant and extensively researched diseases of wheat and other cereal plants [6]. This disease is a major concern worldwide, as it can not only reduce grain quality, but also cause losses in grain yields [7,8,9,10]. In the past, FHB and the resulting yield losses posed a minimal threat to farmers in Lithuania, though the disease has continued to be a severe issue in Lithuanian fields since its first outbreaks were discovered in 2012 [4,11]. Though it is reported that, globally, *F. graminearum* and *F. culmorum* are the main causal agents of FHB, there is also evidence that *F. avenaceum* and *F. poae* can cause fungal infection [3,5]. Species belonging to the genus of *Fusarium* are present everywhere: in soil, air, water, plants and animals. Environmental conditions such as temperature [12,13,14], host resistance [15], humidity [16] and nitrogen fertilization play a key role in the successful spread of FHB infection [17]. However, it was reported that temperature and the duration of wetness of the head are the main factors underlying fungal infection [18]. Depending on the need for ecological conditions, soil-borne *Fusarium* species can be endophytes or pathogens [19]. *Fusarium* spp. survives well as saprophytes on plant debris and can also survive on plant surfaces without causing disease [20]. Mycotoxins produced by fungi of the genus *Fusarium* are very widespread and have great economic importance in terms of their toxicity to animals, humans and other plant pathogens [21]. Among mycotoxin-producing species, the most aggressive and harmful are *F. graminearum* and *F. culmorum*, which synthesize type B trichothecenes such as deoxynivalenol (DON) and its acetyl forms (15-acetyldeoxynivalenol 15ADON and 3-acetyldeoxynivalenol 3ADON) and nivalenol [22,23]. *F. avenaceum* synthesizes beauvericin, enniatins and moniliformin [24,25]. Many mycotoxins remain stable during food processing and are generally resistant to chemical and thermal effects [26,27,28].

Wheat (*Triticum aestivum*), barley (*Hordeum vulgare*), rice (*Oryza sativa*), oats (*Avena sativa)*, rye (*Secale cereale*), triticale (*Triticum secale*) and maize (*Zea mays*) are the main primary hostplants of pathogenic *Fusarium* spp. However, it was found that weeds and wild plants around the field, as well as non-gramineous plants present in agroecosystems, can serve as asymptomatic alternative plants inhabiting FHB-associated *Fusarium* species, thereby increasing disease incidence in associated crop plants [9,29,30,31,32]. In recent decades, scientists have increasingly investigated *Fusarium* spp. residing in weeds and non-gramineous plants. Several assays have shown that asymptomatic and broadleaf weeds, as well as wild grasses, are reservoirs to *Fusarium* spp. related to harmful diseases of gramineous cereals [8,29,32]. Ilic et al. [33] investigated the pathogenicity of thirty isolates (from weeds and plant debris in eastern Croatia) representing 14 *Fusarium* species on wheat and maize seedlings. All tested *Fusarium* spp. isolates were pathogenic to wheat seedlings and the disease index (DI) was statistically significantly higher than the DI compared to the control. The pathogenicity of *Fusarium* isolates for wheat seedlings differed between species and strains. *F. graminearum* isolated from *Amaranthus retroflexus* and *Abutilon theophrasti* were the most pathogenic with a DI of 100.0, while *F. graminearum* from *Chenopodium album*, two isolates of *F. sporotrichioides* from maize debris and *F. avenaceum* from *Agrostemma githago,* were less pathogenic (DIs of 77.5, 76.0, 80.0 and 60.0 respectively). Another study found that isolates of *F. graminearum* from potato and sugar beet cause symptoms of FHB in wheat and produce different mycotoxins in wheat heads and rice grains [34,35]. Mourelos et al. [18] described the isolation of *F. graminearum*, the major causal agent of FHB in Argentina. from florets of healthy weeds belonging to 57 gramineous and non-gramineous asymptomatic species. Fifty-four of the weed species belonging to 19 botanical families were identified as alternative hosts for *F. graminearum*. Dong et al. [36] reported that gramineous weeds harbor the *F. graminearum* species complex that causes FHB in rice. The authors collected 142 weed samples from 10 gramineous weed species. The results showed that the most dominant species from the *F. graminearum* complex was *F. asiaticum*. *Fusarium asiaticum* isolates were able to infect rice and cause FHB on rice heads under greenhouse conditions. Disease severity after 21 DPI ranged from 3% to 30% depending on the isolate. Svitlica et al. [37] investigated the pathogenicity of *F. graminearum* isolated from different plants, including maize, wheat, barley, soybeans, *Arctium lappa* and *Sorghum halepense*. The results showed that the most pathogenic isolate was *F. graminearum* from *Sorghum halepense*. Postic et al. [32] identified 14 *Fusarium* species isolated from 300 isolates belonging to 12 weed families and plant debris. The results showed that *F. graminearum* (20%), *F. verticillioides* (18%), *F. oxysporum* (16%), *F. subglutinans* (13%) and *F. proliferatum* (11%) were present in more than 10% of the population. Other *Fusarium* species such as *F. avenaceum*, *F. concolor*, *F. crookwellense*, *F. equiseti*, *F. semitectum*, *F. solani*, *F. sporotrichioides*, *F. venenatum* and *F. acuminatum* were present at frequencies of < 8%. The results indicated that the most frequently isolated and important species in terms of FHB incidence in Croatia is *F. graminearum*. Recently conducted studies in Lithuania demonstrated that oilseed rape, potatoes, sugar beet, peas and 56 weeds species (all detected in the field) were asymptomatically colonized by nine *Fusarium* species: *F. avenaceum*, *F. culmorum*, *F. graminearum*, *F. equiseti*, *F. tricinctum*, *F. sporotrichioides*, *F. poae*, *F. oxysporum* and *F. redolens* [38,39,40]. In pathogenicity tests, all tested *F. graminearum* (91 isolates) were able to cause FHB symptoms in spring wheat and the disease severity values were comparable to those isolated from the primary hostplants of wheat and barley [40]. The results of a study conducted in 2019 showed that *F. culmorum* isolates from asymptomatic weeds and non-gramineous plants had similar effects to *F. graminearum* on spring wheat. However, the roles of other *Fusarium* spp. in the epidemiology of the FHB remained unclear [41]. Previously, similar findings were published by Pereyra and Dill-Macky, who indicated that *F. graminearum* isolated from *Digitaria sanguinalis* residues caused FHB in wheats [29].

Numerous previous studies showed the extensive adaptations of *Fusarium* species in colonizing the internal tissues of various non-gramineous agroecosystem plants and weeds. However, it is still challenging to fully comprehend the true role of *F. graminearum* and other *Fusarium* species in FHB epidemiology, as morphologically and genetically distinct *Fusarium* species may have high phenotypic diversity. It is, therefore, necessary to investigate the ability of asymptomatically existing *Fusarium* species (especially other than *F. graminearum*) in alternative hosts to cause FHB symptoms in cereals and to determine whether these species are more aggressive to spring wheats. The aim of this study was to evaluate the pathogenicity of *Fusarium avenaceum*, *F. culmorum*, *F. graminearum* and *F. sporotrichioides* isolated from non-gramineous plants and weed species to spring wheat under greenhouse conditions.

## 2. Materials and Methods

### 2.1. Isolation of Fusarium Fungi from Plant Material

This research was conducted in 2021 at the Institute of Agriculture, Lithuanian Research Centre for Agriculture and Forestry (55°23′50″ N; 23°51′40″ E). The presence of *F. avenaceum*, *F. culmorum*, *F. graminearum* and *F. sporotrichioides* was assessed in non-gramineous plants (*Brassica napus*, *Pisum sativum* and *Beta vulgaris*) and in weeds (*Tripleurospermum inodorum*, *Viola arvensis*, *Fallopia convolvulus*, *Capsella bursa-pastoris* and *Poa annua*). The aforementioned plants were collected from cropping system fields, taken to the laboratory, identified and processed for further experiments. Selected visually asymptomatic plants were thoroughly washed, dried, numbered and identified by growth stage and species. *Fusarium* fungi were isolated from all morphological parts of the plant, including roots, crowns, stems, leaves, florets, pods, petioles and fruits. Isolation of *Fusarium* spp. was performed according to Suproniene et al. [39]. Several segments of different parts of the plant (1 cm in size) were sterilized for 3 min in 1% sodium hypochlorite (NaClO) solution and then rinsed 3 times in sterile distilled water (SDW) and dried on sterile filter paper in a laminar. Three segments of different parts of the plant were placed on potato dextrose agar (PDA, Merck) medium and the plates were incubated at 22 ± 2 °C in the dark for 2–4 days. *Fusarium* fungi that appeared were purified via PDA and grown on a Spezieller Nährstoffarmer Agar medium (SNA) for spore mass formation [42].

### 2.2. Identification of Fusarium Fungi

Monosporic cultures were grown on PDA and SNA at 25 ± 2 °C for 10–30 days until the formation of macroconidia. Species were identified based on Nelson et al. [43] and Leslie et al. [44] as descriptors based on visual colonies and microscopic morphological features of conidia typical for each species. During identification a microscope at 10×, 20× and 40× magnification was used to evaluate all possible signs. Once *Fusarium* species were identified, fungal cultures were purified on water agar (WA) media [44]. Three suspension dilutions (10^−1^, 10^−2^ and 10^−3^) were prepared for each identified *Fusarium* isolate. Then, 10 µL of the suspension (10^−3^) was spread in plates with WA and dispersed with a sterile L-shaped spreader. Inverted plates with *Fusarium* cultures on WA were grown in an incubator at 25 ± 2 °C for 2–3 days until the first mono-sporous colonies appeared. Using a microscope, in laminar under aseptic conditions, we determined whether the colony was formed from one conidia or did not come into contact with other conidia. After the assessment, the germinated conidia/monosporic colony was transferred to a PDA medium and incubated at 25 ± 2 °C for 3–7 days until further colony growth. After that, a colony fragment ~5 mm Ø from the peripheral zone was transferred onto an SNA medium and incubated at the same temperature for 10–30 days until the formation of macroconidia.

Confirmation of *Fusarium* spp. identification based on the taxonomic keys [44] was previously described by Suproniene et al. [39,40], who used DNA amplification in PCR assays with species-specific primer pairs reported by Demeke et al. [45] (*F. avenaceum*: J1AF/R; *F. culmorum*: FC01F/R; *F. graminearum*: Fg16F/R; and *F. sporotrichioides*: AF330109CF/R). Sequencing of tef1a (=eEF1a, translation elongation factor 1-a) gene amplicons was carried out for selected strains of *Fusarium* species [39,40]. Among the sequenced *Fusarium* species, only *F. graminearum* strains BN98c, BN425l (from *Brassica napus*), VA153l, VA541s (from *Viola arvensis*), FC144r and FC544r (from *Fallopia convolvulus*) were included in the present study, previously published with codes 98c, 452l, 153l, 541s, 144r and 544r, respectively [40].

### 2.3. Preparation of Spore Suspensions

Spore suspensions were prepared according to Purahong et al. [46] and Suproniene et al. [39,40]. For suspension preparation, *Fusarium* isolates were grown on an SNA medium at a temperature of 22 ± 2 °C for 14–30 days (until they formed a spore mass of macroconidia in sporodochia). For the preparation of spore suspensions, 10 mL of SDW was added to the plate and the spores formed in the micelle and on the surface of the medium were separated by sweeping the surface of the medium with circular movements using an L-shaped spreader. The suspension was filtered through a sterile cotton strainer. The concentration of spores in the suspension was counted using a Neubauer cell counting chamber. A suspension of 1 × 10^5^ spores/mL concentration was used for the wheat inoculation.

### 2.4. Description of Greenhouse Experiment

Experiments included evaluation of the pathogenicity of *F. avenaceum*, *F. culmorum*, *F. graminearum* and *F. sporotrichioides* strains isolated from different alternative hostplants under greenhouse conditions. The pathogenicity of *Fusarium* strains isolated from non-gramineous hostplants to spring wheat was determined according to Purahong et al. [46]. The experiment was divided into two parts: the pathogenicity of *Fusarium* spp. isolates from non-gramineous plants (experiment I) and the pathogenicity of isolates from weeds (experiment II). In experiment I, a total 46 of *Fusarium* isolates, including 35 from non-gramineous plants and 11 from spring wheat, were used. In experiment II, a total of 45 *Fusarium* isolates, including 34 from weeds and 11 from spring wheat, were used. Isolates of *Fusarium* spp. were obtained from the collection of the Microbiology Laboratory, Lithuanian Research Centre for Agriculture and Forestry and were used for floret inoculation. Pots (LxWxH: 13.0 × 8.8 × 11.5 cm) were filled with a commercial pH-adjusted (5.5–6.5) substrate and four spring wheat seeds were planted per pot. The FHB-susceptible spring wheat breeding line “DS-1403-3-DH” was used for pathogenicity tests. The greenhouse was maintained at ±25 °C during the day and ±19 °C at night with a 14-h light and 10-h dark mode. Wheat plants with mineral fertilizer complex (NPK, 11-11-21) were fertilized one week after planting (3 g of fertilizer per pot) and watered 2 times a week. The spikes were inoculated during mid anthesis. Twenty microliters (10 µL/floret) of each *Fusarium* isolate suspension (spore concentration of 1 × 10^5^ spores mL^−1^) and sterile water as a negative control were injected into two adjacent florets at the center of the head (without wounding). The inoculated heads were covered with polyethylene bags for 72 h to ensure the required moisture. The suspension of each isolate was used for the inoculations of 15 heads (3 heads × 5 replicates).

Scheme of experiment I

A total of 47 treatments (Appendix A) were performed, with three (two when missing) isolates (strains) of each *Fusarium* species taken from each plant (3 non-gramineous crop rotation plant species and spring wheat) − (4 *Fusarium* species × 3 isolates × 4 plant species) + 1 negative control (sterile H_2_O) − 3 isolates missing = 47 (Table 1).

Scheme of experiment II

A total of 46 treatments (Appendix A) were performed with two (one or zero when missing) isolates (strains) of each *Fusarium* species taken from each plant (5 weed species and spring wheat) − (4 *Fusarium* species × 2 isolates × 6 plant species) + 1 negative control (sterile H_2_O) − 3 isolates missing = 46 (Table 1).

### 2.5. Analysis of FHB Parameters of Inoculated Spring Wheat

FHB incidence and severity (%) were assessed after the 7th (BBCH 69–71), 14th (BBCH 73) and 21st (BBCH 73–75) days post inoculation (DPI). The area under the disease progress curve (AUDPC) was calculated after the 28th (BBCH 75–77) DPI. All inoculated plants were evaluated. The incidence of disease showed the number of heads affected by the disease, expressed as a percentage. For the assessment of disease severity, we used the visual evaluation scale for the disease developed by Engle et al. [47]. When the grain was fully ripe (BBCH 89), each bundle of 5 heads (replication of each treatment) was cut and packed in a paper bag upon which the date, the treatment of the study and the replication were indicated. The grain was threshed using a laboratory single-ear thresher (Precision Machine model WHTA010002, Co. Inc. Lincoln, NE, USA), cleaned, packed in small paper bags with all the above information and stored in a dry room for no longer than two weeks until the grain analyses. The total number and weight of grains of five heads, biological yield (grain weight per head) and weight of 1000 grains were calculated (TGW).

### 2.6. Statistical Analysis

The statistical analyses were carried out using SAS software package, version 9.3 (SAS Institute Inc., USA) (*p* ≤ 0.05), to identify the significance of pathogenicity between *Fusarium* species obtained from different hostplants. Research data were processed via Tukey’s HSD (honestly significant difference) test (*p* = 0.05). The mean ± SE (standard error of the mean) was used to describe the variability of measurements. The area under the FHB disease progress curve (AUDPC), FHB severity and 1000-grain weight (TGW) were statistically evaluated, and calculations were performed in Microsoft Office Excel 2007. This program was also used to present the data graphically.

## 3. Results

### 3.1. Evaluation of FHB Severity, AUDPC and TGW under Wheat Inoculation with Fusarium Fungi Isolated from the Non-Gramineous Plants

Overall, 43 out of 46 of *Fusarium* spp. isolates obtained from non-gramineous plants present in the crop rotation fields were confirmed to be pathogenic to spring wheat according to the ability to cause FHB symptoms during the floret inoculation test in the greenhouse. After 21 DPI, in *Fusarium* spp.-inoculated heads, the FHB severity ranged from 9.3% to 69.6% and was significantly higher (*p* < 0.01) compared to the water control (4.3%) (Figure 1). In *F. avenaceum*-inoculated heads, the FHB severity values varied from 9.3% to 19.0% (on average, 12.6%). The isolate BN19c from *B. napus* showed the highest FHB severity (19%), while the isolates BV33.3s from *B. vulgaris* and PS10fl from *P. sativum* were the least pathogenic (7.8% and 8.0%) and did not differ from the water control. In *F. culmorum*-inoculated heads, FHB severity values ranged from 9.5% to 45.7% (on average, 31.8%), with the highest value (45.7%) observed in isolate BV15.1l from *B. vulgaris*. In *F. graminearum*-inoculated heads, the FHB severity values varied from 11.7% to 69.6% (on average, 26.8%). The isolate 5PS3p3–1 from *P. sativum* showed the highest FHB severity value (69.6%). In *F. sporotrichioides*-inoculated heads, the FHB severity values ranged from 9.7% to 16.7% (on average, 11.9%). The highest FHB severity (16.7%) showed isolate 9SWSP17 from spring wheat. Isolate PS37s from *P. sativum* was the least pathogenic (8.2%) among the *F. sporotrichioides* isolates and did not differ from the water control.

After 28 DPI, in *Fusarium*-inoculated heads, the AUDPC varied from 161.5% to 1044.6% and was significantly higher (*p* < 0.01) compared to the water control (70.9%) (Figure 2). In *F. avenaceum*-inoculated heads, the AUDPC values ranged from 161.5% to 331.6% (on average, 254.8%). The isolate BN19c from *B. napus* showed the highest AUDPC value (331.6%). In *F. culmorum*-inoculated heads, the AUDPC values ranged from 332.5% to 683.7% (on average, 480.9%), with the highest value (683.7%) observed in isolate 8SW5SP2 from spring wheat. In *F. graminearum*-inoculated heads, the AUDPC values ranged from 191.5% to 1044.6% (on average, 438.2), with the highest value (1044.6%) observed in isolate 5PS3p3-1 from *P. sativum*. In *F. sporotrichioides*-inoculated heads, the AUDPC values ranged from 189.0% to 291.8% (on average, 220.7%), with the highest value (291.8%) observed in isolate 9VKV17 from spring wheat.

At full ripening stage (BBCH 89), *F. culmorum* isolates BN26r and BN39fl from *B. napus* (26.3 g and 29.2 g, respectively); isolates BV15.1l and BV142.1pe from *B. vulgaris* (29.7 g and 28.5 g, respectively); and isolates SW4SP11, 8SW5SP2 and 8SW1SP3 (30.6 g, 31.9 g and 30.2 g, respectively) from spring wheat showed significantly (*p* < 0.01) lower TGW values compared to the water control (41.7 g) (Figure 3). In *F. graminearum*-inoculated heads, isolate 5PS3p3-1 from *P. sativum* showed a significantly (*p* < 0.01) lower (29.2 g) TGW compared to the water control. In *F. sporotrichioides*-inoculated heads, isolate 8SW5SP19 from spring wheat showed a significantly (*p* < 0.01) (32.5 g) lower TGW compared to water. In *F. avenaceum* inoculated-heads, lower 1000 grain weight values were detected, but the differences were not significant.

### 3.2. Evaluation of FHB Severity, AUDPC and TGW under Wheat Inoculation with Fusarium Fungi Isolated from Weeds

After 21 DPI, in *Fusarium*-inoculated heads, the FHB severity ranged from 6.2% to 81.0% and was significantly higher (*p* < 0.01) compared to the water control (1.9%) (Figure 4). In *F. avenaceum*-inoculated heads, the FHB severity varied from 6.2% to 19.3% (on average, 10.5%). Isolate 1 FC1178fl from *F. convolvulus* showed the highest FHB severity value (19.3%). In *F. culmorum*-inoculated heads, the FHB severity ranged from 13.0% to 81.0% (on average, 46.4%). Isolate CBP1401r from *C. bursa-pastoris* showed the highest FHB severity value (81.0%). In *F. graminearum*-inoculated heads, FHB severity values ranged from 7.3% to 26.2% (on average, 13.2%). Isolate PA1130c from *P. annua* showed the highest FHB severity value (26.2%). In *F. sporotrichioides*-inoculated heads, the FHB severity values varied from 7.1% to 11.4% (on average, 8.9%). Isolate 9SWSP17 from spring wheat showed the highest FHB severity value (11.4%).

After 28 DPI, in *Fusarium*-inoculated heads, the AUDPC values ranged from 134.2% to 1206.6% and were significantly (*p* < 0.01) higher compared to the water control (87.8%) (Figure 5). In *F. avenaceum*-inoculated heads, the AUDPC values varied from 163.8% to 346.4% (on average, 241.8%). Isolate FC1178fl from *F. convolvulus* showed the highest AUDPC value (346.4%). In *F. culmorum*-inoculated heads, the AUDPC values ranged from 174.8% to 1206.6% (on average, 751.3%). Isolate CBP1401r from *C. bursa-pastoris* showed the highest (1206.6%) AUDPC value. In *F. graminearum*-inoculated heads, AUDPC values ranged from 154.0% to 403.3% (on average, 227.4%). Isolate PA1130c from *P. annua* showed the highest (403.3%) AUDPC value. Isolate 6SW4SP1 from spring wheat showed a significantly lower (*p* < 0.01) AUDPC value (129.3%) and did not differ from the water control. In *F. sporotrichioides*-inoculated heads, the AUDPC values varied from 134.2% to 177.6% (on average, 150.7%). Isolate 8SW5SP19 from spring wheat showed the highest AUDPC value (177.6%) but did not differ from the water control.

At the full ripening stage (BBCH 89), *F. culmorum* isolates CBP1147c and CBP1401r from *C. bursa-pastoris* (27.5 and 19.0 g), PA1129c and PA1129f from *P. annua* (24.7 and 27.7 g) and FC1088r from *F. convolvulus* (29.2 g) showed significantly (*p* < 0.01) lower TGW values compared to the water control (39.7 g) (Figure 6). The *F. graminearum* isolates PA1130c from *P. annua* (30.0 g) and 6SW5SP1 from spring wheat (32.2 g) showed a significantly lower (*p* < 0.01) 1000 grain weight compared to the water control.

### 3.3. Comparison of the Pathogenicity of Fusarium Fungi Isolated from Different Hostplants

In this study, the aggressiveness of different *Fusarium* species isolated from different hostplant groups was compared (weeds, non-gramineous plants and *Triticum*). The results showed that FHB severity was highest (46%) when spring wheat was inoculated with a *F. culmorum* species group isolated from weeds; similar results were obtained for the species isolated from wheat (42%). Meanwhile, *F. graminearum* and *F. culmorum* isolates from non-gramineous plants caused FHB severity in wheat of 27% and 29%, respectively. These data indicate that *Fusarium* isolates from various hostplants can produce different disease severities. Although *F. graminearum* is noted to be the most pathogenic among *Fusarium* species for wheat, in our case, the conditions were more favorable for the development of *F. culmorum*. *F. avenaceum* and *F. sporotrichioides* isolates, which caused disease with similar severity (8–12%) in all hostplant groups (Figure 7).

Significant differences were observed between groups of *Fusarium* strains in the AUDPC values. *F. culmorum* strains isolated from weeds showed significantly higher AUDPC values (*p* < 0.01) compared to other *Fusarium* species and the water control under greenhouse conditions (Figure 8). In the non-gramineous hostplant group, *F. culmorum* and *F. graminearum* strains showed significantly higher AUDPC values (*p* < 0.01) compared to other *Fusarium* species and the water control. In the *Triticum* group, *F. culmorum* strains were also the most aggressive and showed significantly higher AUDPC values (*p* < 0.01) compared to other *Fusarium* species and the water control.

The thousand grain weight (TGW) values between wheat inoculated with *Fusarium* isolates obtained from different hostplants (weeds, non-gramineous plants and *Triticum*) were found to be very similar (Figure 9). A statistically significant decrease in TGW was observed when wheats were inoculated with *F. culmorum* isolated from weeds, non-gramineous plants and *Triticum* (on average, decreases of 25.4%, 23.3% and 21.8% respectively), compared to the water control. *F. avenaceum*, *F. graminearum* and *F. sporotrichioides* reduced TGW compared to the control, but the results were statistically insignificant.

## 4. Discussion

Although numerous studies have already investigated the epidemiology of *Fusarium* species, there is still a lack of information about how alternative cropping system plants and weeds contribute to the spread of FHB. The present study reports the results of the pathogenicity of *Fusarium* spp. isolates from asymptomatic non-gramineous plants and weeds to spring wheat under greenhouse conditions. As previously reported, weeds and non-gramineous plants can serve as alternative hosts for *Fusarium* species [9,18,32,33,37,39,40], leading cereals to become contaminated by pathogenic fungi. Our first experiment with non-gramineous cropping system plants confirmed that species such as *B. napus*, *P. sativum* and *B. vulgaris* can harbor FHB-associated *Fusarium* fungi such as *F. avenaceum*, *F. culmorum*, *F. graminearum* and *F. sporotrichioides*. These findings agree with the previous study of Rasiukeviciute et al. [38], which presented non-cereal plants as alternative hostplants. Isolates from the above-mentioned plants produced FHB symptoms in the tested spring wheat inoculated with the spore suspension (1 × 10^5^) and some were more pathogenic compared to those isolated from primary hostplants. *F. avenaceum* isolate BN19c from *B. napus*, *F. culmorum* isolate BV15.1l from *B. vulgaris* and *F. graminearum* isolate 5PS3p3-1 from *Pisum sativum* showed the highest FHB severity (%). Our results indicated that FHB severity differs between isolates and between species. Additionally, *Fusarium* species also play a key role in the infection of FHB, as *F. graminearum* from non-gramineous plants was most pathogenic. These findings are consistent with other study results [33,48,49]. A study conducted under field conditions by Matelionienė et al. [48] showed that *F. graminearum* isolates from both wheat and weeds cause severe FHB disease, whereas *F. avenaceum* species did not show heavy disease symptoms. Notably, other crucial factor for pathogenicity to wheat include the production of different mycotoxins by *Fusarium* species. A previous study by Janaviciene et al. [50] investigated mycotoxin concentrations in spring wheat inoculated with *F. graminearum* strains isolated from weeds, including *P. annua*, *T. inodorum*, *V. arvensis*, *F. convolvulus*, *B. napus* and *T. aestivum*. The authors observed that the levels of mycotoxin production depended not only on the trichothecene genotype but also (and mostly) on the strain and environmental conditions. In addition to other studies showing that *F. graminearum* from non-cereal plants is pathogen to cereals under field conditions [38], our findings also illustrate that *F. graminearum* (5PS3p3-1) from *P. sativum* can cause severe FHB symptoms. The aforementioned isolate showed the highest FHB severity and highest AUDPC value (%) under greenhouse conditions. It is also evident, that *F. culmorum* isolates from non-gramineous plants were as pathogenic as *F. graminearum* isolates, since isolate BV15.1 l from *B. vulgaris* showed higher FHB severity (%) than isolates from wheat. In addition, our results indicate that *F. culmorum* isolates BN26r and BN39fl from *B. napus*, isolates BV15.1l and BV142.1pe from *B. vulgaris* decreased the 1000 grain weight by 37%, 30%, 28.8% and 31.8%, respectively, compared to the water control. These findings agree with those of Brennan et al. [12], who showed that *F. culmorum* and *F. graminearum* strains caused a losses of 54.3% and 46.9%, respectively, to 1000 grain weight in wheat cultivars. Our results also confirm that although *F. avenaceum* is reported to cause FHB symptoms in cereals, it is not as pathogenic as *F. culmorum* and *F. graminearum*. There is, however, very little information about the pathogenicity of *F. sporotrichioides* to spring wheat. Our study showed that *F. sporotrichioides* isolates from non-gramineous plants were able to cause FHB-associated symptoms, although the pathogenicity of the isolates was less strong than that of other *Fusarium* strains.

Our study also agrees with the findings of other scientists who reported that non-symptomatic weeds can harbor FHB-associated *Fusarium* fungi [18,39,40]. In our second experiment, *F. avenaceum*, *F. culmorum*, *F. graminearum* and *F. sporotrichioides* were observed in weeds such as *F. convolvulus*. *C. bursa pastoris*, *P. annua*, *T. inodorum* and *V. arvensis* without any visible symptoms. Suproniene et al. [39,40] reported that *F. graminearum* is one the most colonizing and pathogenic *Fusarium* species among weeds along with *F. culmorum*. Our study also indicated that these two *Fusarium* species are the most pathogenic. Specifically, spring wheat inoculated with isolates from these species showed the highest levels of FHB severity, while the AUDPC and 1000 grain weight were the lowest. Among all *Fusarium* spp. isolates from weeds, the *F. culmorum* isolate CBP1401r from *C. bursa-pastoris* was the most pathogenic and showed the highest FHB severity and AUDPC value. This isolate decreased the 1000 grain weight by 52% compared to the water control. These findings agree with previous study by Jenkinson and Parry [49], who reported that *C. bursa-pastoris* was among the weed hostplants that harbor *F. avenaceum* and caused FHB symptoms in winter wheat within the UK. The *F. avenaceum* isolate FC1178fl from *F. convolvulus*, *F. culmorum* isolate CBP1401r from *C. bursa-pastoris* and *F. graminearum* isolate PA1130c from *P. annua* showed highest FHB severity and were more pathogenic to spring wheat compared to isolates from the primary hostplant—wheat. These results show that not only primary hostplants, but also non-symptomatic weeds can harbor pathogenic *Fusarium* fungi and produce FHB symptoms in spring wheat. Moreover, isolates from *F. avenaceum* and *F. sporotrichioides* did not have any significant difference in 1000 grain weight and were less pathogenic than the other two *Fusarium* species. These findings are similar to those reported in the study by M. Gerling et al. [51], which demonstrated that weeds were 15 times more strongly infected with *Fusarium* fungi than herbaceous plants. The difference in the hostplant colonization of fungi is explained by the greater genetic diversity of weeds than cultivated plants. Due to this diversity, weeds are less susceptible to diseases themselves and do not show any signs of disease associated with FHB. Linde et al. [52] also confirmed that pathogens from genetically diverse hosts, such as weeds, may be more virulent than pathogens from monocultural hosts.

In our research, we also evaluated the aggressiveness of different *Fusarium* species isolated from various hostplants. We determined that, under greenhouse conditions, *F. culmorum* strains from weeds and primary hostplant *Triticum* were more pathogenic than the other three investigated *Fusarium* species. Additionally, in non-gramineous plants, *F. culmorum* was the most pathogenic alongside *F. graminearum*, which is already known to be one of the most aggressive *Fusarium* strains. Therefore, our findings relating to *F. culmorum* are very important as they show not only that *Fusarium* strains from *F. graminearum* complex can cause FHB-associated symptoms, but also that *F. culmorum* should be considered one of the most aggressive *Fusarium* strains against spring wheat. Despite this factor, it remains necessary to further investigate other *Fusarium* species isolated from alternative hostplants and their ability to cause FHB. We should also obtain more information on which species contributes most to the spread of pathogenic disease.

## 5. Conclusions

In conclusion, our results indicate that asymptomatic non-gramineous crop rotation plants such as *Brassica napus*, *Pisum sativum* and *Beta vulgaris* and weed species such as *Tripleurospermum inodorum*, *Fallopia convolvulus*, *Poa annua*, *Capsella bursa-pastoris* and *Viola arvensis*, which are found in cereal-based crop rotations, can serve as reservoirs of pathogenic *Fusarium* species. In the present study, we showed that isolates from these plants can cause FHB symptoms in spring wheat under greenhouse conditions. We determined that *F. culmorum* strains isolated from weeds and *Triticum aestivum* were more pathogenic to spring wheat, whereas in non-gramineous plant groups *F. culmorum* and *F. graminearum* were the most pathogenic to spring wheat. A significant thousand grain weight reduction was caused only by *F. culmorum* strains isolated from all hostplant groups. Despite these results, further research is necessary to investigate the significance and role of other *Fusarium* species from alternative plants as such species play a key role in FHB epidemiology.

## Figures and Tables

**Figure 1 pathogens-11-01467-f001:**
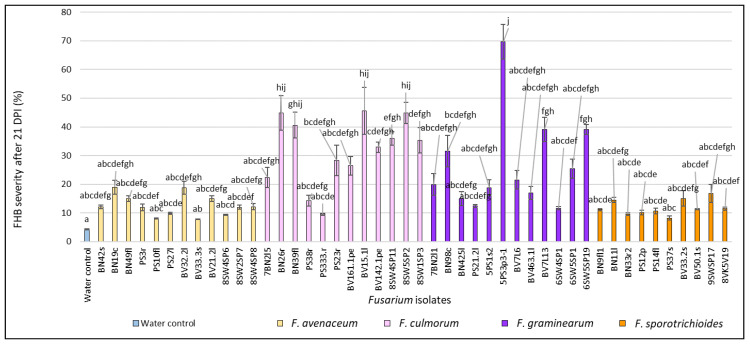
FHB severity (%) in spring wheat heads inoculated with *Fusarium* isolates isolated from non-gramineous plants at 21 DPI. Spring wheat florets were inoculated with the spore suspension (1 × 10^5^) in the middle of anthesis. FHB severity (%) values are the means of 5 replicates ± standard error (SE). Different letters above the bars indicate significant differences according to Tukey’s HSD (honestly significant difference) test (confidence level = 0.95) compared to the water control (*p* < 0.01). BN—*Brassica napus*, PS—*Pisum sativum*, BV—*Beta vulgaris*, SW—spring wheat, c—crown, f—fruit, fl—flower, l—leave, r—root, p—pod, pe—petiole, s—stem, sp—head.

**Figure 2 pathogens-11-01467-f002:**
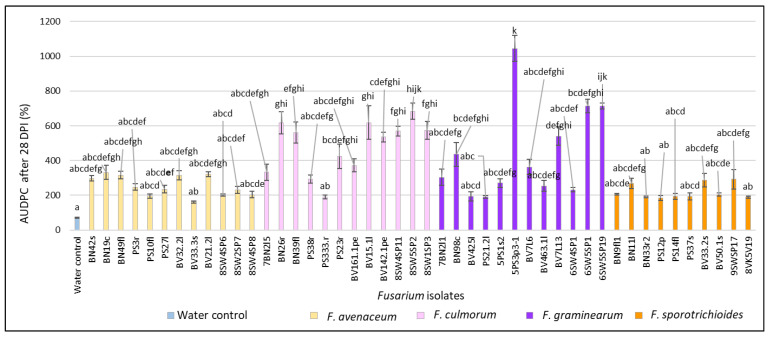
AUDPC values (%) in spring wheat inoculated with *Fusarium* isolates isolated from non-gramineous plants at 28 DPI. Spring wheat florets were inoculated with the spore suspension (1 × 10^5^) in the middle of anthesis. AUDPC values are the means of 5 replicates ± standard error (SE). Different letters above the bars indicate significant differences according to Tukey’s HSD (honestly significant difference) test (confidence level = 0.95) compared to the water control (*p* < 0.01). BN—*Brassica napus*, PS—*Pisum sativum*, BV—*Beta vulgaris*, SW—spring wheat, c—crown, f—fruit, fl—flower, l—leave, r—root, p—pod, pe—petiole, s—stem, sp—head.

**Figure 3 pathogens-11-01467-f003:**
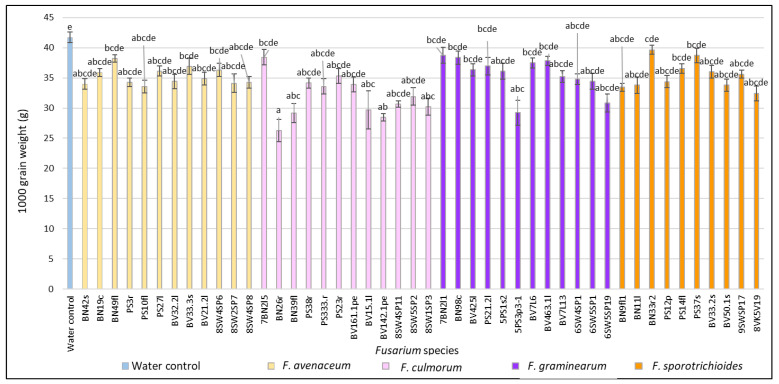
1000 grain weight (g) (TGW) in spring wheat inoculated with *Fusarium* isolates isolated from non-gramineous plants at the full ripening stage. Spring wheat florets were inoculated with the spore suspension (1 × 10^5^) in the middle of anthesis. Here, 1000 grain weight values are the means of 5 replicates ± standard error (SE). Different letters above the bars indicate significant differences according to Tukey’s HSD (honestly significant difference) test (confidence level = 0.95) compared to the water control (*p* < 0.01). BN—*Brassica napus*, PS—*Pisum sativum*, BV—*Beta vulgaris*, SW—spring wheat, c—crown, f—fruit, fl—flower, l—leave, r—root, p—pod, pe—petiole, s—stem, sp—head.

**Figure 4 pathogens-11-01467-f004:**
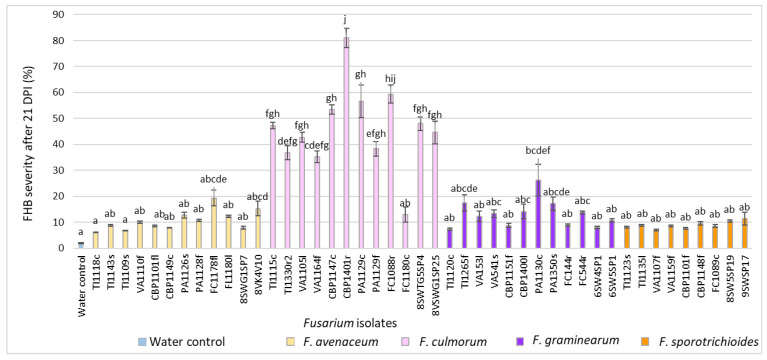
FHB severity (%) in spring wheat heads inoculated with *Fusarium* isolates isolated from weeds at 21 DPI. Spring wheat florets were inoculated with the spore suspension (1 × 10^5^) in the middle of anthesis. FHB severity (%) values are the means of 5 replicates ± standard error (SE). Different letters above the bars indicate significant differences according to Tukey’s HSD (honestly significant difference) test (confidence level = 0.95) compared to the water control (*p* < 0.01). TI–*Tripleurospermum* inodorum, VA–*Viola arvensis*, CBP–*Capsella bursa-pastoris*, PA–*Poa annua*, FC–*Fallopia convolvulus*, SW–spring wheat, G–glyphosate soil, c–crown, f–fruit, fl–flower, l–leave, r–root, s–stem, sp–head.

**Figure 5 pathogens-11-01467-f005:**
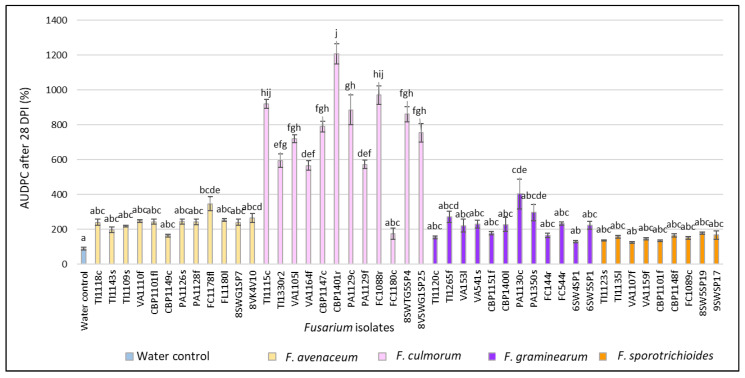
AUDPC values (%) in spring wheat inoculated with *Fusarium* isolates isolated from weeds at 28 DPI. Spring wheat florets were inoculated with the spore suspension (1 × 10^5^) in the middle of anthesis. AUDPC values are the means of 5 replicates ± standard error (SE). Different letters above the bars indicate significant differences according to Tukey’s HSD (honestly significant difference) test (confidence level = 0.95) compared to the water control (*p* < 0.01). TI—*Tripleurospermum*
*inodorum*, VA—*Viola arvensis*, CBP—*Capsella bursa-pastoris*, PA—*Poa* a*nnua*, FC—*Fallopia convolvulus*, SW—spring wheat, G—glyphosate soil, c—crown, f—fruit, fl—flower, l—leave, r—root, s—stem, sp—head.

**Figure 6 pathogens-11-01467-f006:**
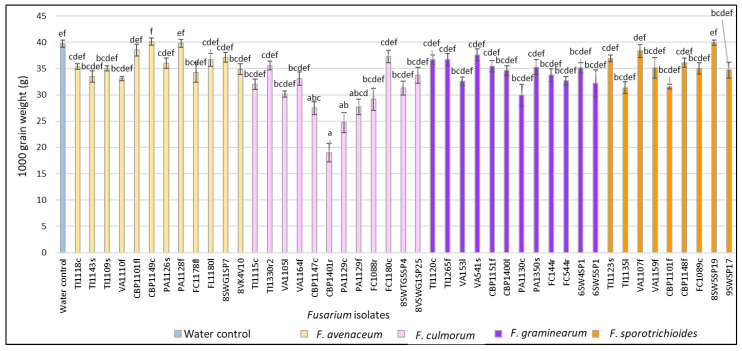
The 1000 grain weight (g) (TGW) in spring wheat inoculated with *Fusarium* isolates isolated from weeds at the full ripening stage. Spring wheat floret values are the means of 5 replicates ± standard error (SE). Different letters above the bars indicate significant differences according to Tukey’s HSD (honestly significant difference) test (confidence level = 0.95) compared to the water control (*p* < 0.01). TI—*Tripleurospermum inodorum*, VA—*Viola arvensis*, CBP—*Capsella bursa-pastoris*, PA—*Poa* a*nnua*, FC—*Fallopia convolvulus*, SW—spring wheat, G—glyphosate soil, c—crown, f—fruit, fl—flower, l—leave, r—root, s—stem, sp—head.

**Figure 7 pathogens-11-01467-f007:**
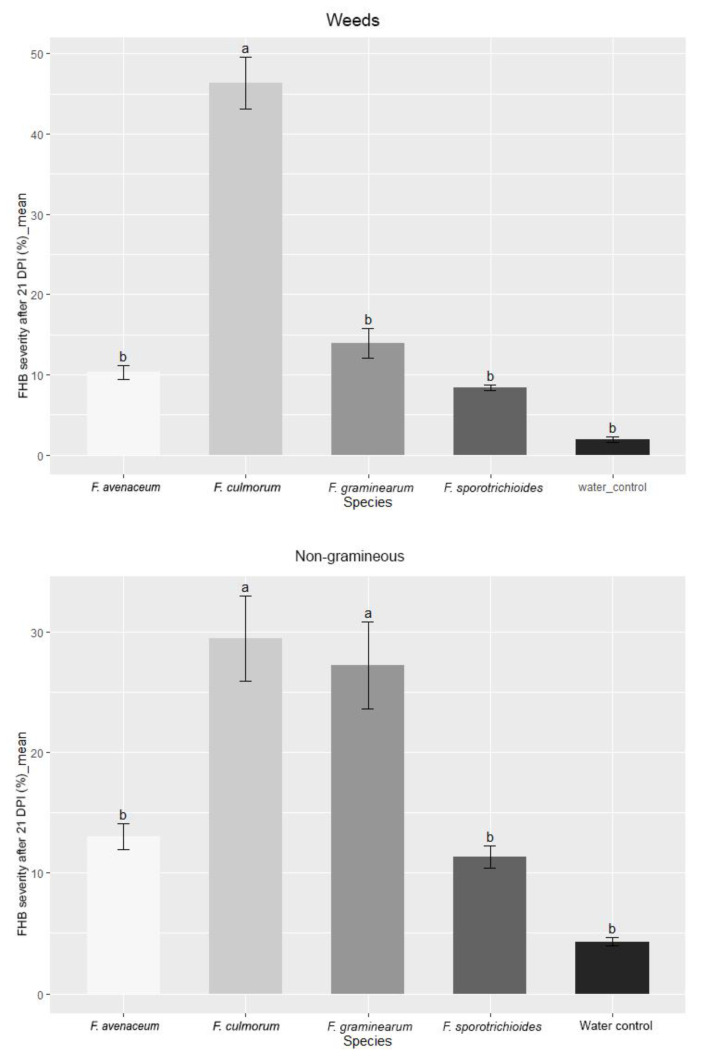
FHB severity (%) in spring wheat inoculated with *Fusarium* species isolated from different hostplants. FHB severity values are the means of all isolates of one *Fusarium* species ± standard error (SE). Different letters above the bars indicate significant differences according to Tukey’s HSD (honestly significant difference) test (confidence level = 0.95).

**Figure 8 pathogens-11-01467-f008:**
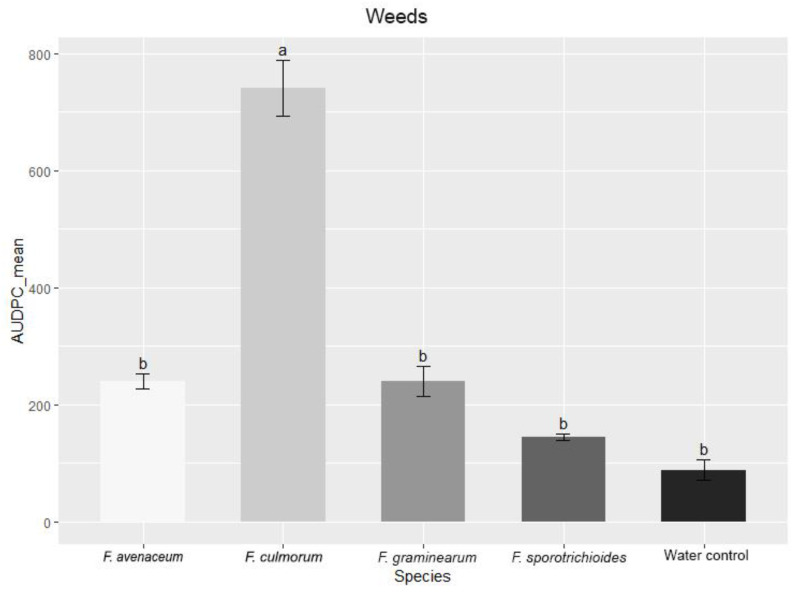
AUDPC values (%) in spring wheat inoculated with *Fusarium* species isolated from different hostplants. AUDPC values are the means of all isolates of one *Fusarium* species ± standard error (SE). Different letters above the bars indicate significant differences according to Tukey’s HSD (honestly significant difference) test (confidence level = 0.95).

**Figure 9 pathogens-11-01467-f009:**
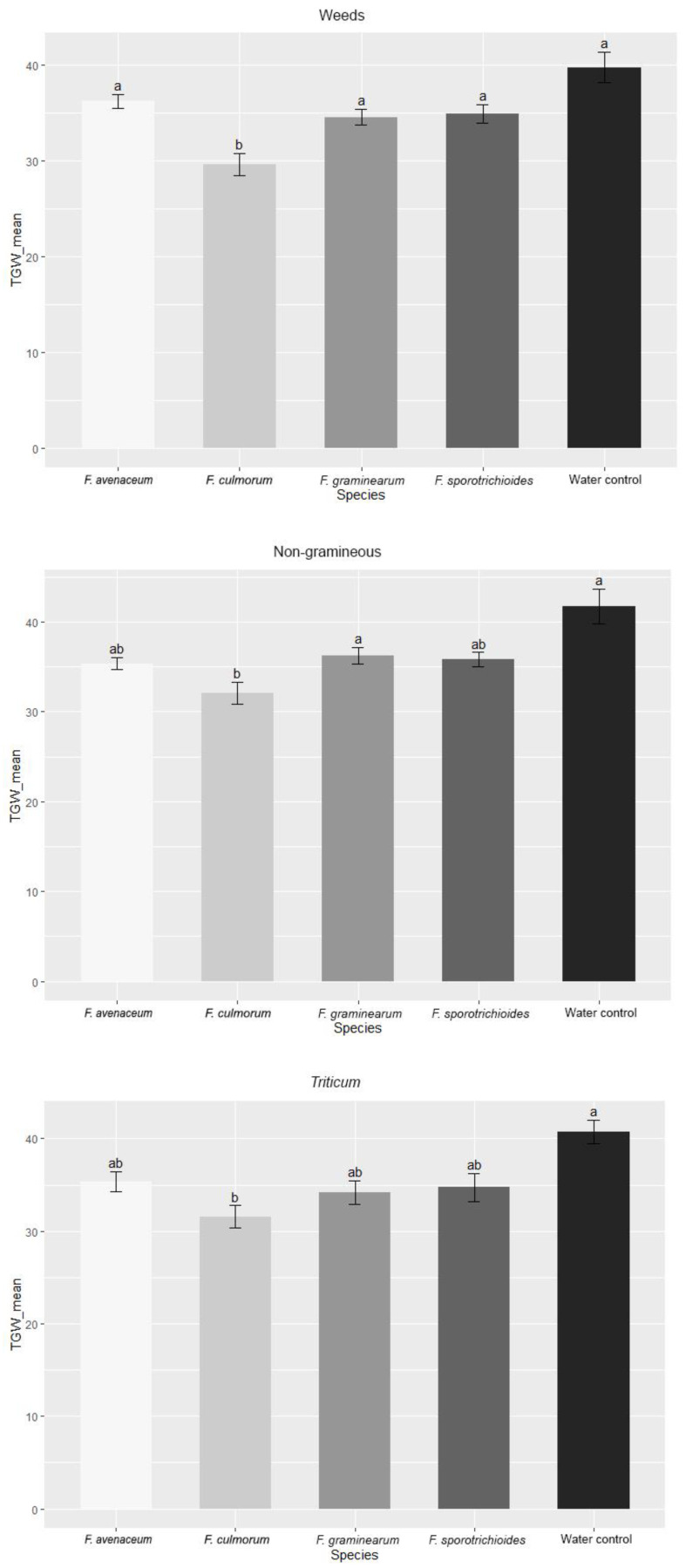
TGW values (g) in spring wheat inoculated with *Fusarium* species isolated from different hostplants. TGW values are the means of all isolates from one *Fusarium* species ± standard error (SE). Different letters above the bars indicate significant differences according to Tukey’s HSD (honestly significant difference) test (confidence level = 0.95).

**Table 1 pathogens-11-01467-t001:** Information on *Fusarium* isolates selected for spring wheat inoculation in the greenhouse experiments.

Experiment	Isolates per *Fusarium* Species	Isolate Number per Hostplant
I experiment	*F. avenaceum*—12	*T. aestivum*—11
	*F. culmorum*—12	*B. napus*—12
	*F. graminearum*—12	*P. sativum*—12
	*F. sporotrichioides*—10	*B. vulgaris*—11
Total in I	46 isolates	46 isolates
II experiment	*F. avenaceum*—12	*T. aestivum*—8
	*F. culmorum*—12	*V. arvensis*—8
	*F. graminearum*—12	*C. bursa pastoris*—8
	*F. sporotrichioides*—9	*P. annua*—6
		*F. convolvulus*—7
		*T. inodorum*—8
Total in II	45 isolates	45 isolates

## Data Availability

Not applicable.

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
