# Peer review of "Pathogenicity of Asymptomatically Residing Fusarium Species in Non-Gramineous Plants and Weeds to Spring Wheat under Greenhouse Conditions"

_pathogens, 2022, doi:10.3390/pathogens11121467_

Round 1

Reviewer 1 Report

This manuscript brings data on the pathogenicity of four Fusarium species derived from non-gramineous plants and weed species to spring wheat under greenhouse conditions. The manuscript is well written and interesting for readers - phytopathologists. Some minor corrections are needed:

Abstract: The scientific names of plant species that appear for the first time in the text should not be abbreviated (line 24: Brassica napus, Beta vulgaris; line 27: Capsella bursa-pastoris).

In Materials and Methods, the name and version of the software used for statistical analyses are not mentioned.

Check the number of fungal isolates throughout the text because it does not match the data in the tables. Furthermore, the data in Table 1 do not match the data in the Appendix Tables. According to the data in the Appendix tables, 46 Fusarium isolates were used in experiment I, and 45 isolates were used.

Line 204: Correct a total number of isolates.

Line 205: Correct a total number of isolates.

Lines 222-232: Check the data in both experiment schemes.

Table 1: In experiment I, correct the data on the number of isolates (F. sporotrichioides – 10; B. vulgaris – 11). In experiment II, add the missing “Tripleurospermum inodorum – 8” and correct the total number of isolates to 45.

Check and correct the table numbers because they do not match.

line 223 “App. A Table 1”  vs  line 464 “Appendix A Table 3”

line 229 “App. B Table 2”  vs  line 469 “Appendix B Table 4”

It would be better to number the Appendix tables as Table S1 and Table S2 (Supplementary Material).

Despite the fact that sterile water was injected into the wheat heads as a negative control, 4.3% FHB severity was found in the wheat heads. What species of Fusarium caused the infection of the control variant?

Line 369: Cite Figure 7C at the end of the sentence.

Page 12, Figure 7C: The graph is lacking a negative control.

The chapters Supplementary Materials, Funding, Acknowledgements, Author Contributions, Conflict of Interest, and so on are missing at the end of the manuscript.

Reviewer 2 Report

This manuscript focused on the comparative evaluation of pathogenicity of different Fusarium species that isolated from small-grain cereals, non-gramineous plants and weeds. This topic is very interesting for study. Previously, the authors demonstrated that different mono- and dicotyledons in agroecosystems are potential fungal infection reservoirs. In this research, the inoculation of spring wheat with 91 isolates of four Fusarium species, including isolates obtained from weeds and non-gramineous plants, was performed in green-house conditions, and after the different quantitative parameters of FHB were analyzed.

Data of research in submitted manuscript are original and quite informative, but the presentation and discussion of results can be improved.

Extensive editing of English language and style of presented manuscript required.

Some of specific comments on the manuscript as following indicated:

Abstract

Please, describe more clearly the novelty and significance of obtained results. The authors should summarize the significant findings, including the comparison of pathogenicity of four species of Fusarium fungi isolated from different host plants.

Line 13. Please, correct “cereal” on “cereals”.

Line 14. Please, rephrase the sentence part. For example, “but it has recently been discovered that asymptomatic non-gramineous plants and weeds can serve as alternative source for fungi associated with FHB”.

Line 17. Please, change “derived” on “isolated”.

Line 24, 27. Please, give full names of plants.

Line 26. Please, delete “(%)” in both cases, and add “from” after “range”, “varied”. Also, it possible to use “in the range //-//”.

Last 28. Please, change “weed species” on “weeds” and “survey” on “study”.

Line 29. Please, delete “fields”.

Line 28-29. In total, last sentence do not contain the specific conclusion based on the experimental scheme and data analysis.  It should be changed or deleted.

Line 31. Keywords duplicated the title. Please, choose other words.

Introduction

Line 57. Please, correct “enniatins”

Line 71. Please, add abbreviation “(DI)” after “disease index”

Lines 73, 75. Please, use “DI” instead of “disease index”.

Line 96. Please, correct “F. crookwellense

Materials and Methods

Line 124. Please, correct the title of 2.1 section on “Isolation of Fusarium fungi from plant material”

Line 128. Please, delete “L.” after “Brassica napus” or add the names of authors for all plants.

Line 173. Please, use “SDW” instead of “sterile distilled water”.

Lines 178-179. Please, correct sentence on “The suspension of 1x105 spores/mL concentration was used for the wheat inoculation”.

Line 180. Please, change the order of sections 2.5 and 2.4.

In first should be 2.4 “Description of greenhouse experiments”, after 2.5 “Analysis of FHB parameters of inoculated spring wheat”.

Line 193. What is the reason for milling and frozen of grain samples?

It will be interesting to analyses other parameters characterizing the relationship between pathogens and host, for example, to evaluate the levels of fungal DNA and mycotoxin in grain.

Line 234. Please, rename Table 1. For example “Table 1. Information on Fusarium isolates selected for spring wheat inoculation in greenhouse experiments”

Results

As a suggestion: the results can be divided on three parts:

3.1 Evaluation of FHB severity, AUDPC and TGW under wheat inoculation with Fusarium fungi isolated from the non-gramineous plants

3.2 Evaluation of FHB severity, AUDPC and TGW under wheat inoculation with Fusarium fungi isolated from the weeds

3.3 Comparison of pathogenicity of Fusarium fungi isolated from different host plants

Please, colorize the columns for clear visibility in Figures 1-6.

Lines 237, 299. Please, change “derived” on “isolated”.

Lines 360-361. Please, delete the information in “()”. It will be more informative, if “weeds, non-gramineous, Triticum” add directly on graphs of Fig. 7 instead “A,B,C”, respectively.

The Latin names of fungi in fig.7 should be in italic.

Why authors did not present summarizing data belong to FHB severity and TGW are the same as for AUDPC? It also should be demonstrated as figures or tables.

Discussion

The authors should revise this part and provide more references to support their findings.

Please, pay attention to next articles:

https://doi.org/10.3390/agronomy12040823

https://doi.org/10.1128/aem.02177-21

Please, correct reference list as recommended by the ACS style guide.

Round 2

Reviewer 2 Report

Authors have addressed my comments and made the changes requested. I have no further objections.

The revised manuscript now contains all necessary information. Overall, it is interesting study, which can be continued.

A few comments that do not affect the manuscript content:

Lint 94. Please, change “F. asiaticum isolates” on “Fusarium asiaticum isolates” or “Isolates of this species”.

Line 112. Please, add the space after ““.

Line 113. Please, delete word “primary”.

Line 369. “The” should not be bold.

Line 381. Please, change “wheats” on “wheat” and delete “Triticum” after “42%”

Line 382. Please, clarify “isolates from non-gramineous plants caused disease severity” on “isolates from non-gramineous plants caused FHB severity in wheat”

Line 383. Please, change first “different” on “various”

Lines 384-385. Please, delete “the species” before “F. graminearum” and add “among Fusarium species” after “pathogenic”.

Lines 414-415. Please, clarify “between hostplants” on “between wheat inoculated with Fusarium isolates obtained from different hostplants”

Lines 416-417. Please, delete “species”, change “obtained’ on “isolated” and delete “groups”.

Line 524. Please, change “Triticum” on “Triticum aestivum”.

Lines 548, 553. As suggestion: delete the decoding under the table and add in the tables the column “Plant tissue” contains information on the parts of plants from which the fungi were isolated. The columns of “Host Plants” and “Latin name” also can be combined in the one, since their content is essentially the same.
